# Cryo-EM structure of renal AL amyloid fibrils from a patient with λ1 light chain amyloidosis

Chenyue Yu[1,2,3,7], Yeyang Ma[4,5,7], Heng Li[6], Kai Liu[1,2,3], He Huang [1,2,3] ✉ &
Kun Zhao [1,2,3] ✉

Systemic light-chain amyloidosis (AL) is characterized by the misfolding and aggregation of immunoglobulin light chains (LCs) into amyloid fibrils, leading to multiorgan deposition and dysfunction, with the kidneys being one of the most commonly involved organs. Here, we report high-resolution cryo-electron microscopy (cryo-EM) structures of AL amyloid fibrils from the kidney of a male patient with renal AL amyloidosis. Two distinct polymorphic fibril structures, polymorph A and polymorph B, were identified, both featuring ordered cores (Gln16-Ser95) with β-sheet-rich architectures stabilized by interchain hydrogen bonds and salt bridges. Notably, six mutations in the IGLV1-44*01 gene sequence, including Gln39His and Tyr37Phe, were identified within the fibril core. These mutations influence fibril stability and aggregation by altering intramolecular and intermolecular interactions, such as CH-π stacking and salt bridge formation. Comparative analysis with previously reported heart-derived IGLV1-44 fibrils reveals structural variations linked to light-chain sequence differences. Our findings provide critical insights into the molecular determinants of fibril assembly and organ tropism in AL amyloidosis.

Systemic light-chain amyloidosis (AL) is a disorder characterized by the abnormal aggregation of immunoglobulin light chains (LCs) into oligomers and amyloid fibrils, which subsequently deposit in various organs and tissues, leading to progressive organ dysfunction. The disease is associated with clonal plasma cell dyscrasia[1]. As of 2018, an estimated 73,567 cases had been reported globally over the preceding two decades, with the prevalence continuing to rise over time[2]. The heart and the kidney are the most frequently affected organs in systemic AL amyloidosis, although virtually all organs—except the brain—can be involved[3]. Renal involvement typically manifests as heavy proteinuria or nephrotic syndrome; however, the lag between the

characteristic pathological changes and the onset of clinical symptoms contributes to an early diagnosis rate of less than 30%[4]. The prognostic situation of the patient is primarily determined by the extent of myocardial amyloid infiltration, with a current median overall survival of 12–18 months[5]. These clinical challenges underscore the urgent need for advances in understanding the molecular pathogenesis of the disease and the mechanisms underlying organ-tropism amyloid deposition.

Among all types of systemic amyloidosis, AL amyloidosis exhibits the greatest clinical heterogeneity. This variability arises from the genetic recombination of variable (V), joining (J), and constant (C) gene

[1]Bone Marrow Transplantation Center of the First Affiliated Hospital & Liangzhu Laboratory, Zhejiang University School of Medicine, Hangzhou, China. [2]Institute of Hematology, Zhejiang University, Hangzhou, China. [3]Zhejiang Province Engineering Research Center for Stem Cell and Immunity Therapy, Hangzhou, China. [4]National Key Laboratory for Development and Utilization of Forest Food Resources, Zhejiang A&F University, Hangzhou, China. [5]Provincial Key Laboratory for Non-wood Forest and Quality Control and Utilization of Its Products, Zhejiang A&F University, Hangzhou, China. [6]Kidney Disease Center of The First Affiliated Hospital, Zhejiang University School of Medicine, Hangzhou, China. [7]These authors contributed equally: Chenyue Yu, Yeyang Ma. ✉e-mail: huanghe@zju.edu.cn; kunzhao@zju.edu.cn

segments, somatic hypermutation, and junctional diversity at the *V/J* interface during *V/J* recombination[6–8]. As a result, the LC sequences derived from either the λ or κ loci are nearly unique to each individual patient[9,10]. Previous studies have shown that LCs assemble into homodimers, with each monomer comprising two immunoglobulin domains. The N-terminal variable domain (VL), in particular, exhibits a high degree of sequence variability[11,12]. The polymorphic amyloid fibril proteins predominantly originate from the VL domain of the LC precursor[13]. Somatic mutations in the immunoglobulin lambda variable (IGLV) gene segments, which encode the LC variable domains, reduce the thermodynamic stability of the protein, thereby facilitating amyloid fibril formation[1,14].

IGLV1-44 is a germline gene of the immunoglobulin lambda (λ) LC variable (*V*) region, primarily functioning in immune responses. It encodes a protein component of the immunoglobulin complex, typically localized to the extracellular space and plasma membrane. As an LC gene, IGLV1-44 plays a critical role in generating antibody diversity, enabling immunoglobulins to recognize and bind a wide range of antigens. Previous studies have demonstrated a significant association between IGLV1-44 and cardiac AL amyloidosis, with this gene being markedly overrepresented among λ LCs implicated in cardiac involvement, compared to other germline genes[15,16]. This association suggests that IGLV1-44 may have a unique role in amyloid fibril formation, possibly due to mutations that reduce protein stability and thereby promote amyloidogenic propensity[17]. More recent studies, however, have shown that IGLV1-44 is also the most frequently used IGLV subfamily among patients with predominant renal involvement. This indicates a potential genetic predisposition or pathophysiological mechanism linking IGLV1-44 to renal AL amyloidosis. Structural features or specific amino acid sequences encoded by this gene may render the LCs more prone to misfolding and aggregation, facilitating amyloid fibril deposition within renal tissue[18].

Amyloid deposits are typically persistent and resistant to degradation, making structural elucidation of amyloid fibrils a critical avenue for understanding their in vivo aggregation mechanisms and guiding therapeutic development. In general, AL fibrils exhibit the characteristic structural features of amyloid fibrils, such as the cross-β architecture and an average width of approximately 15 nm[13,19]. However, since amyloid fibrils often derive from natively folded proteins under physiological conditions, structures of fibrils generated in vitro under denaturing conditions may not fully represent those found in patient-derived amyloids[20,21]. Cryo-electron microscopy (cryo-EM) combined with three-dimensional (3D) reconstruction has emerged as the preferred method for determining the molecular structure of ex vivo amyloid fibrils. To date, the structures of fibrils extracted from the hearts of five patients with advanced cardiac amyloidosis have been reported. These include λ6-AL55, λ3-FOR005, λ1-FOR001, λ1-FOR006, FOR103, FOR010, and λ3-AL59, each exhibiting distinct folding conformations[17,22–26]. Notably, fibrils extracted from both the heart and the kidney of the same patient (AL55) display nearly identical folding patterns, highlighting the immunoglobulin LC sequence as a key determinant of fibril structure[27]. It is important to emphasize that although the kidney is one of the most frequently affected organs in AL amyloidosis—with involvement observed in 60–70% of cases[28]—there is still a lack of systematic structural characterization of amyloid fibrils specifically from patients with isolated renal involvement. Current understanding of AL fibril structure remains limited in scope across different patient subgroups with organ-specific involvement, underscoring the need for broader and more targeted structural studies.

In this study, we employed cryo-EM to determine the high-resolution 3D structure of LC amyloid fibrils extracted from the kidney tissue of a patient with AL amyloidosis. Negative-stain transmission electron microscopy (TEM) analysis revealed the presence of two distinct structural polymorphs, designated as polymorph A and polymorph B, with corresponding cryo-EM reconstructions reaching resolutions of 2.9 Å and 3.0 Å, respectively. The fibril widths were measured at approximately 13 nm and 14 nm for polymorphs A and B. Further structural analysis demonstrated that the fibril core is composed of the Glu16-Ser95 segment of the LC protein. This region assembles into a highly ordered protofilament architecture through antiparallel β-sheet stacking, forming the structural backbone of the fibrils.

## Results

### Extraction and sequence analysis of amyloid fibrils from a patient's renal biopsy

To investigate the structural organization of native amyloid fibrils, we selected a 66-year-old male patient diagnosed with renal AL amyloidosis and nephrotic syndrome as the study subject. Renal tissue samples were obtained via percutaneous kidney biopsy, and AL amyloid fibrils were extracted from the tissue using an optimized protocol for isolating tissue-derived amyloid fibrils[19]. To analyze the molecular composition of the isolated fibrils, we employed a proteomics-based approach. Sodium dodecyl sulfate–polyacrylamide gel electrophoresis was used to separate the fibrillar proteins, followed by liquid chromatography–tandem mass spectrometry of the gel bands to accurately determine the amino acid sequence characteristics of the fibril-forming proteins (Supplementary Fig. 1). Previous studies have demonstrated that the immunoglobulin light chain variable region (IGLV) is associated with organ tropism[29]. Bioinformatic analysis revealed that the core component of the deposited fibrils in this patient corresponds to a germline-derived sequence encoded by the IGLV1-44*01 gene, consistent with earlier findings linking IGLV1-44 fragments to predominant renal involvement[18]. Compared to the native germline sequence, the fibril protein exhibited six mutations within the IGLV1-44 region—Ser26Thr, Ser31Gly, Tyr37Phe, Gln39His, Gly65Ala, and Asp86His—likely arising from B cell clone-specific somatic hypermutation[30]. These mutations may reduce the stability of the LC and thereby promote its misfolding, facilitating amyloid fibril formation and aggregation.

### Cryo-EM structural determination of the renal AL fibril

To investigate the structure of the extracted fibrils, we first performed negative-staining TEM. This initial analysis revealed the presence of fibrils with the characteristic helical symmetry of amyloid but was insufficient to resolve high-resolution details (Supplementary Fig. 2). To determine the atomic structures, we subsequently employed cryo-EM. Following cryo-EM data collection, we used RELION for helical reconstruction of this fibril species. 3D classification from helical image processing of 5786 filaments revealed two major polymorphs of twisted fibrils, herein designated as polymorphs A and B (Supplementary Fig. 3). Based on the 0.143 Fourier shell correlation (FSC) criterion, the reconstructions were refined to spatial resolutions of 2.9 Å for polymorph A and 3.0 Å for polymorph B (Supplementary Fig. 4). The density maps showed that each fibril consists of two protofilaments (Fig. 1a, b). Consistent with the density map, polymorph A fibril displays a (pseudo) $2_1$ screw symmetry, characterized by a helical twist of −179.58° and a rise of 2.46 Å. Polymorph B exhibits C1 helical symmetry, characterized by a helical twist of 0.87° and a helical rise of 4.92 Å for the winding dimer (SI Table 1). Both polymorphs form right-handed helices, with polymorph A measuring approximately 14 nm in width and a half-pitch of ~106 nm, while polymorph B has a width of ~13 nm and a half-pitch of ~102 nm. The ordered fibril core extends from Gln16 to Ser95 and contains an intramolecular disulfide bond formed between Cys22 and Cys89 (Fig. 2a).

### Salt bridges and hydrophobic interactions mediate stability in the fibril protein fold

The 3D cryo-EM map was fitted with contiguous peptide segments corresponding to residues Gln16-Ser95 of the IGLV1-44 fibril protein

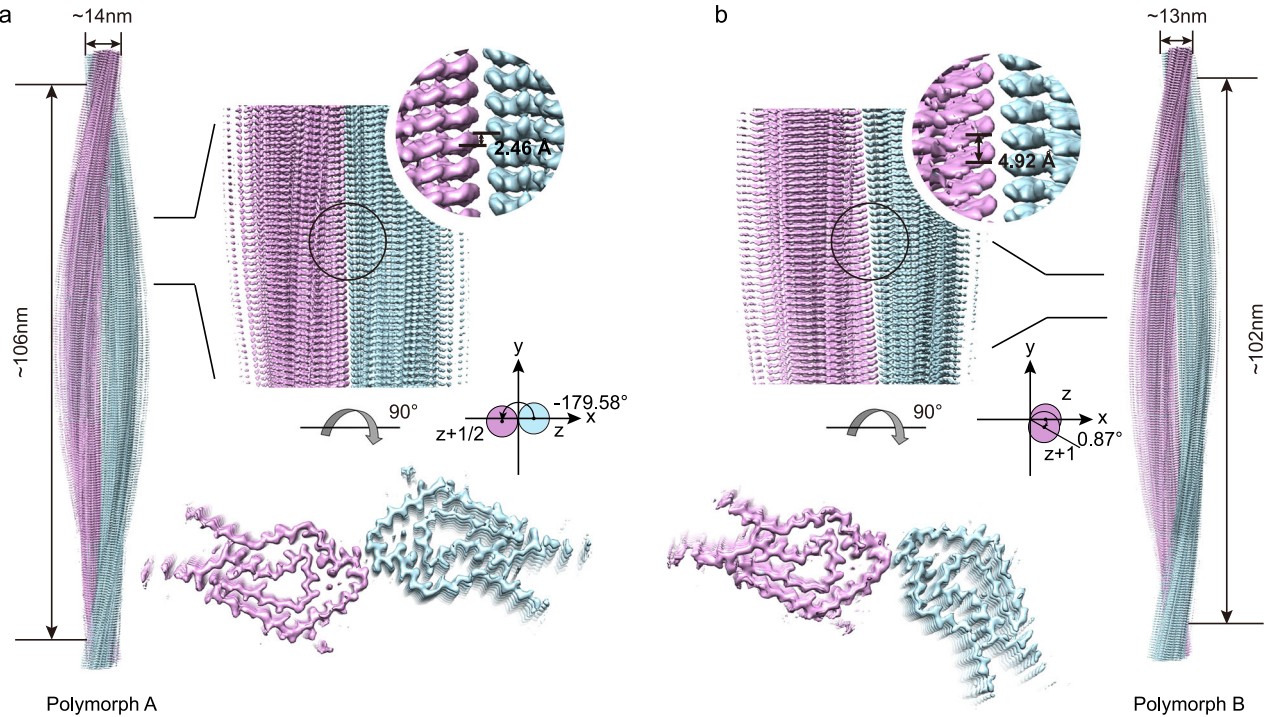

**Fig. 1 | Cryo-EM 3D reconstruction density map of the IGLV1-44 fibril.** Cryo-EM density maps of the polymorph A (**a**) and polymorph B (**b**). Fibril width, length of half-pitch (180° helical turn), helical rise, and twist angle are indicated. The twist angle is graphically illustrated. Protofilaments are colored individually.

(Fig. 2b, c). The N- and C-termini of the protein were surrounded by diffuse density (Supplementary Fig. 3), indicating structural disorder in the first and last -15 residues (Fig. 2c). Within the fibril core, β-strand conformations were identified at residues Gln16–Ser21, Thr26–Gly30, Asn32–Asn35, Gly42–Tyr50, Asn53–Val59, Ser64–Ser71, Ser73–Ile76, Glu84–Tyr88, and Ala90–Asp93, which we designated as β1 through β9, respectively (Fig. 2c, d). These β-strands interact along the fibril z-axis via backbone hydrogen bonding and side-chain interactions, including aromatic stacking and polar residue contacts.

Both polymorphs A and B are composed of two protofilaments entwined together; however, the protofilament interaction interfaces differ significantly. Polymorph A's interface is stabilized by two pairs of inter-chain aromatic residue interactions, specifically a symmetric CH-π stacking between His39-Pro41 and Pro41-His39, markedly enhancing interface stability (Fig. 2a). In contrast, polymorph B maintains protofilament interactions through a single CH-π interaction between His39 and Pro41, with an interface topology exhibiting asymmetric features. Notably, both polymorphs display an identical β-sheet layering pattern within the core of individual protofilaments, which is underscored by a very low Cα RMSD of 0.001 Å between the two protofilaments. This indicates that structural variations arise from differences in inter-protofilament interface regulation rather than conformational rearrangements within protofilaments.

The fibril surface exhibits a pronounced charge distribution pattern, with a molecular surface highly enriched in charged and polar amino acid residues. The N-terminal region is dominated by positive electrostatic potential (blue), whereas the C-terminal region exhibits a negative potential (red) (Fig. 3a). Protein folding creates two major cavities, labeled A and B. Cavity A's inner surface is primarily composed of hydrophobic side chains such as Leu, Ile, and Phe, conferring pronounced hydrophobicity; in contrast, cavity B is enriched with polar residues like Ser and charged side chains including Asp and Lys, resulting in a hydrophilic character (Fig. 3a, b). Notably, compared to cardiac fibrils (Supplementary Fig. 5a, b), our kidney-derived fibrils display a more abundant and complex surface charge pattern, which is

likely an adaptation to the charge-rich microenvironment of the kidney.

**Implication of patient-specific mutations for fibril morphology**
Compared to the germline IGLV1-44 amino acid sequence, the amyloidogenic LC contains six mutations (Fig. 3c, d), which are widely recognized as drivers of amyloidosis in affected patients[31,32]. Among these, the Gln39His mutation introduces a charge polarization effect via the histidine side chain, enabling a cross-chain CH-π stacking interaction with the adjacent Pro41 residue of a neighboring protofilament subunit. This interaction facilitates the formation of a unique fibril interface, leading to a dimer architecture distinct from other amyloid fibrils. It thereby constitutes a core structural basis for IGLV1-44 fibril assembly and enhances the aggregation kinetics of the protein subunits compared to their native or non-amyloidogenic states. The Tyr37Phe mutation (Fig. 3b) may strengthen the van der Waals forces and hydrophobic synergism between β-sheet layers of the fibril, further stabilizing the fibril. Additionally, the Asp86His mutation (Fig. 3b) can form a salt bridge with Glu84, reinforcing the internal fibril architecture. In contrast, the Ser26Thr, Ser31Gly, and Gly65Ala mutations appear to have minimal impact on fibril structure. Collectively, these distinct mutation types may contribute to organ-tropism deposition patterns, such as cardiac or renal involvement. Detecting mutation profiles in patient LCs could therefore guide the selection of targeted therapies, providing a rationale for first-line treatment options in clinical practice. For instance, if a mutation profile closely aligns with that characteristic of cardiac-derived fibrils, a cardiotropic treatment strategy would be prioritized. Conversely, the detection of a mutation profile associated with renal amyloidosis would justify a renally targeted approach. However, it is important to note that the existing maps of organ-specific mutational signatures remain incomplete. This evolving framework for mutational profiling nonetheless allows for a more personalized and biologically rational selection of first-line therapies, ultimately aiming to improve treatment outcomes.

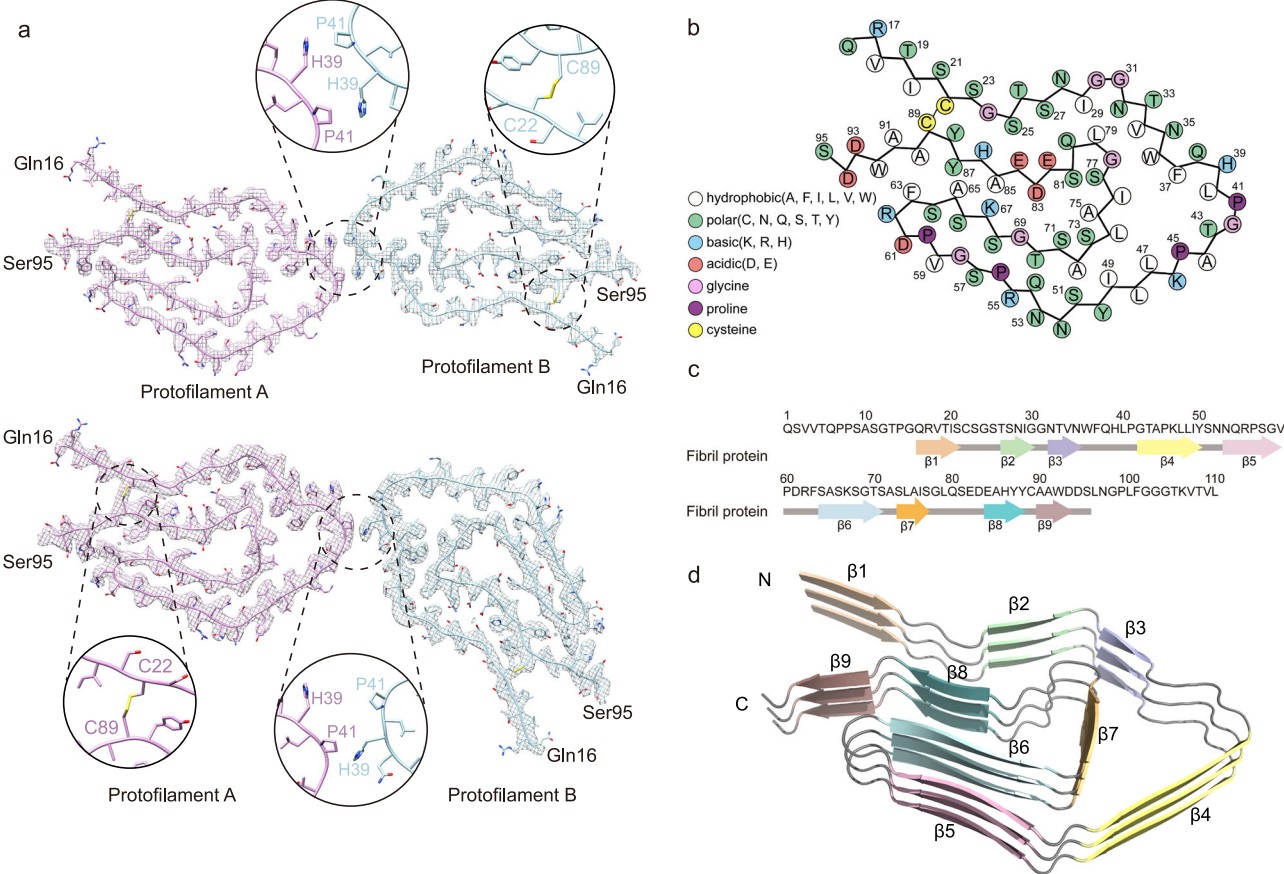

**Fig. 2 | Cryo-EM structure of the IGLV1-44 fibril. a** Density maps with structural models for polymorph A (Upper) and polymorph B (Lower). Two dimers are shown and colored by molecules. The two intertwining protofilaments are colored in plum and light blue, respectively. Molecular interfaces are shown in the zoom-in views with involved residues and interactions labeled. **b** Schematic representation of the fibril protein packing. **c** Schematic representation of the secondary structure of the fibril protein. Arrows indicate β-strands. Continuous lines indicate ordered conformation. **d** Ribbon diagram of a stack of three fibril proteins. The β-strands are numbered and labeled in different colors.

## Comparative analysis with cardiac fibrils and native light chain structures

The fibril structure exhibits significant differences from the native folded VL domain. Although both structures are rich in β-sheets, critical conformational rearrangements were observed around the disulfide bond region, which covalently links the N- and C-terminal regions of the protein. In the native conformation, these two domains adopt a parallel N→C orientation, whereas in the fibril assembly, they switch to an antiparallel topological arrangement (Supplementary Fig. 6). This conformational transition is attributed to an intramolecular domain rotation occurring during protein misfolding, specifically involving a 180° topological inversion of one domain around the disulfide bond axis.

A previously reported cardiac-derived AL fibril structure[26], also belonging to the IGLV1-44 family, shows that hydrophobic interactions between the C-terminal extended Leu96 and N-terminal Val18 bring the protein's N- and C-termini into close proximity, forming a stable hydrophobic core cavity together with the intramolecular disulfide bond (Fig. 4a). In contrast, our study reveals that the renal-derived amyloid fibril displays diffuse density beyond residue Ser95, resulting in a bifurcated conformation of the N- and C-termini (Fig. 4b). Further structural analysis delineates the mechanisms by which the Gln39His and Tyr37Phe mutations reshape the fibril core: Gln39His mediates inter-protofilament contacts, while Tyr37Phe causes an inward flipping of the residue. This dual rearrangement culminates in the formation of a larger internal cavity specific to the renal fibril.

Notably, the Asp86His mutation in the renal fibril forms a stable intermolecular salt bridge with the neighboring Glu84 residue, significantly enhancing fibril stability; whereas in the cardiac fibril, structural stabilization is mediated by a salt bridge interaction between Asp86 and Arg25. These findings reveal pronounced molecular assembly differences between AL amyloid fibrils derived from distinct tissues (Fig. 4c), establishing a structural basis for how mutation-driven fibril polymorphism contributes to organ tropism.

## Discussion

In this study, we report the cryo-EM structure of ex vivo fibrils isolated from a patient with renal AL amyloidosis and nephrotic syndrome. Distinct from previously published fibril protein structures, the renal AL amyloid fibrils exhibit unique topological features. Native LC can form protofilaments under the combined effects of somatic mutations, co-factors, and the local tissue microenvironment. When stable intermolecular interactions exist between protofilaments, two protofilaments can further associate into a dimeric structure, and may even assemble into trimers or higher-order aggregates. Variations in the intermolecular forces can also lead to the formation of distinct polymorphs. The 3D reconstructed fibril is composed of two protofilaments entwined together, wherein the somatically mutated His39 residue forms a critical interaction interface with the Pro41 residue of an adjacent fibril subunit. The segment spanning Gln16 to Ser95 adopts a stable spatial conformation, whereas the first and last 15 amino acid residues are disordered and unresolved in our structure. The fibril core

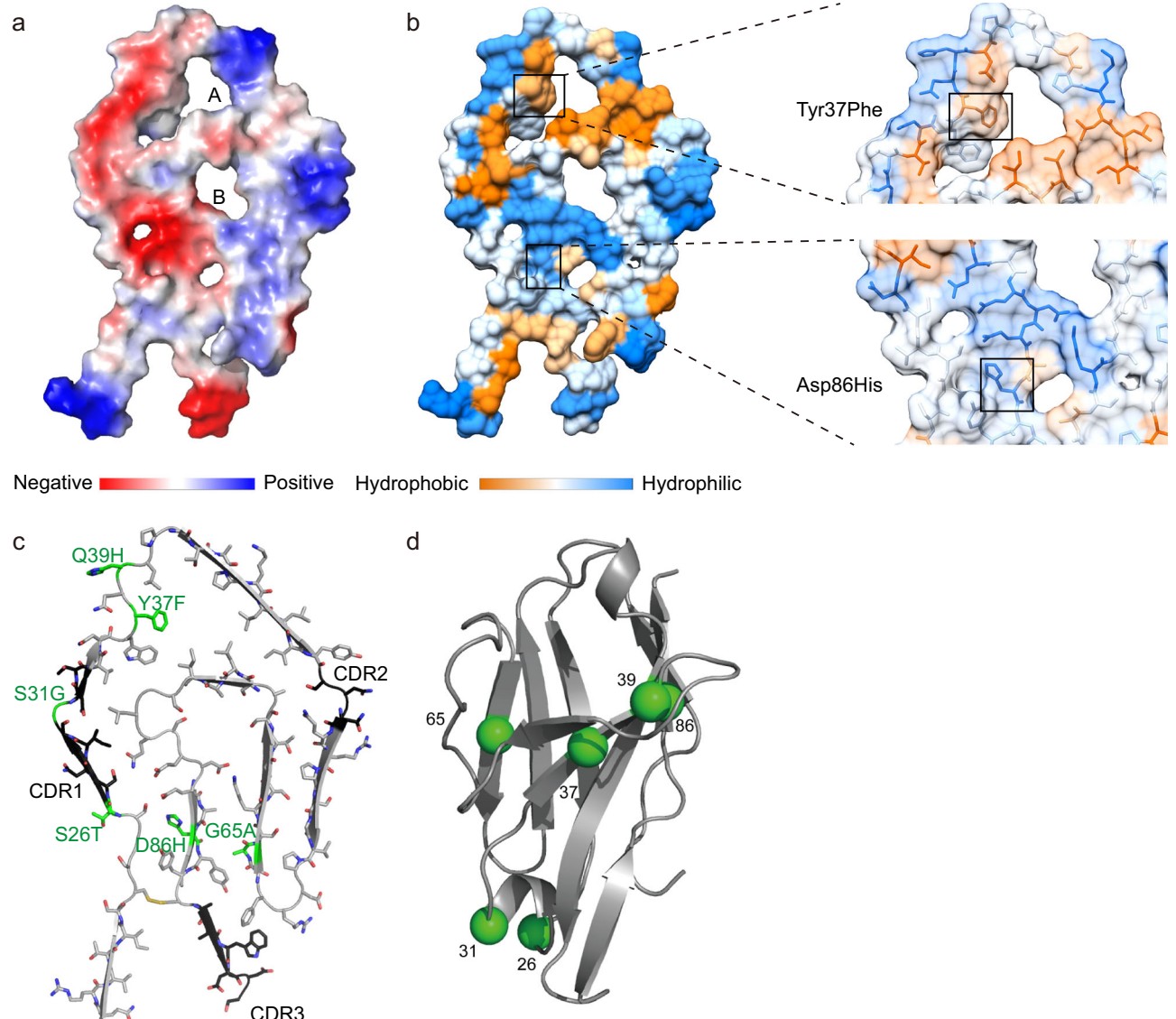

**Fig. 3 | Surface properties and mutations of the IGLV1-44 fibril. a** Electrostatic surface representation of the fibril protein. Red indicates a negative charge, blue positive, and white neutral. **b** Hydrophobic surface representation of the fibril protein. Mutant residues Tyr37Phe and Asp86His, with their local environments, are shown in the zoom-in view. Orange indicates hydrophobic residues, blue hydrophilic, and white neutral. **c** Fibril protein with CDRs (black) and mutations (green) highlighted. **d** Location of mutations in the corresponding, natively folded LC VL domains that are based on the GL segments IGLV1-44*01(PDB entry 1BJM). Mutations are colored in green.

displays a periodic cross-β arrangement along the fibril axis, a spatial organization that may confer mechanical elasticity under stress.

Systemic AL amyloidosis represents a clinically heterogeneous group of diseases[5,33]. Investigating the conserved structural features of amyloid fibrils across different patients and affected organs can help elucidate key molecular events that are highly preserved during pathological conformational conversion, thus providing crucial structural insights into the molecular pathways of protein misfolding cascades. Integrating current advances in amyloid fibril structural studies[17,22–26], several commonalities emerge: fibril cores are typically formed by the VL domain of precursor λ-LCs, although emerging evidence demonstrates that the constant domain (CL)[22] can also be incorporated into the core structure in some cases. However, relative to the native state, fibril proteins consistently exhibit an antiparallel N-C orientation around the disulfide bond, indicating global structural rearrangements and/or unfolding occur during the conversion of native LC or LC fragments into fibrils. Notably, all previously reported fibrils consist of single protofilaments, whereas the renal-derived fibrils

we resolved contain two protofilaments: polymorph A fibril displays a (pseudo) $2_1$ screw symmetry, while polymorph B displays C1 helical symmetry.

Previous studies have demonstrated that IGLV6-57 is a frequently expressed germline gene in patients with LC amyloidosis[34], and its encoded archetypal LC protein mediates multiorgan amyloid deposition. AL55 LCs isolated from cardiac and renal tissues of the same individual undergo pathological aggregation independently in their respective microenvironments, yet maintain significant structural conservation[25,27]. In contrast, the renal-derived amyloid fibrils belonging to the IGLV1-44 gene family analyzed here share overall structural similarity with cardiac-derived fibrils but reveal marked structural heterogeneity upon cryo-EM comparison. This heterogeneity arises from key amino acid substitutions induced by germline mutations and the consequent remodeling of intermolecular interaction networks, suggesting that amyloid fibrils evolve adaptive organotropism through microenvironment-driven structural modulation, ultimately resulting in differential organ

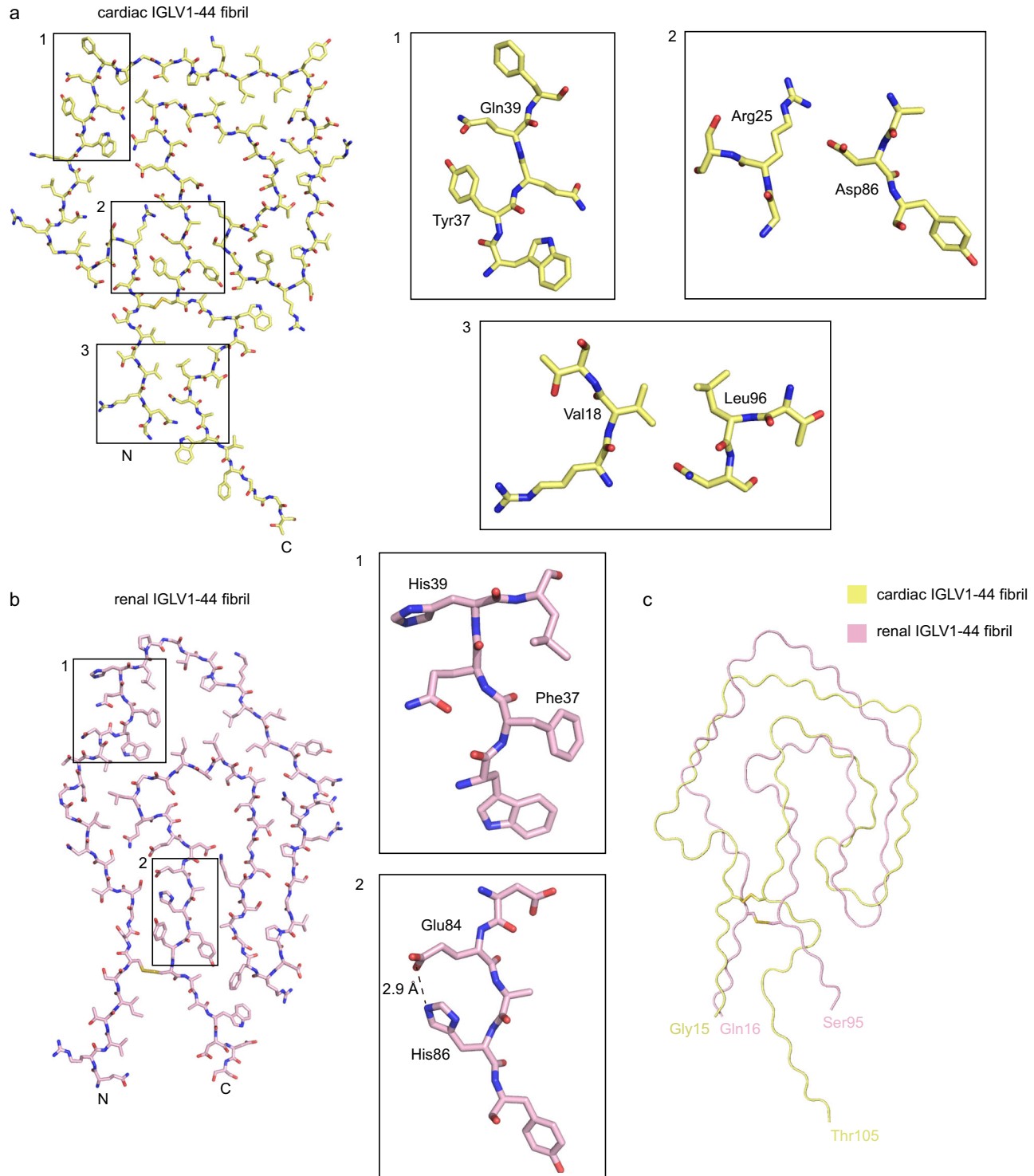

**Fig. 4 | Comparison of the cryo-EM structures of cardiac IGLV1-44 fibril and renal IGLV1-44 fibril. a** Cryo-EM structure of the cardiac IGLV1-44 fibril. The fibril is shown in pale yellow. A magnified view of the boxed region highlights key residues in a schematic representation, with residue numbers labeled. **b** Cryo-EM structure of the renal IGLV1-44 fibril. The fibril is shown in light pink. A magnified view of the boxed region highlights key residues in a schematic representation, with residue numbers labeled. **c** A structural alignment of the renal (colored in light pink) and heart (colored in pale yellow) IGLV1-44 fibrils.

deposition. For example, the renal fibrils in our study display pronounced surface charge polarity that may electrostatically complement the high ionic strength of the renal interstitial milieu, whereas the cardiac fibrils from the same family exhibit weakened surface charge distribution, potentially adapting to the low shear extracellular matrix topology of the myocardium[35]. Different tissues and organs exhibit significant variations in their extracellular matrix architecture, 3D topology, interstitial fluid composition, fluid shear forces[36], and cell surface proteins[37], all of which likely influence amyloidogenesis. We propose that both the tissue microenvironment and LC sequence contribute to the structural diversity of amyloid fibrils to varying degrees.

## Methods

### Fibril extraction from renal

Renal tissue from a 66-year-old male patient with nephrotic syndrome and AL lambda amyloidosis was obtained from the First Affiliated Hospital of Zhejiang University School of Medicine. Biological sex was obtained from medical records. Because the study involved only a single clinical sample, sex-disaggregated analysis was not possible and not applicable. Tissue was stored frozen (−80 °C) without fixation until use. This stored tissue has been fully consumed for the experiments described herein. During routine diagnostic procedures, renal biopsy samples were collected via percutaneous renal needle biopsy following acquisition of informed consent for research-oriented specimen preservation and experimental applications. The presence of amyloid deposits was evaluated by Congo red staining analysis under polarized light and by electron microscopy. Amyloid typing was confirmed by immuno-electron microscopy[38]. Organ involvement was defined according to international criteria[39]. Fibrils were extracted as described by Annamalai et al.[19]. In brief, 250 mg of tissue were diced and washed five times with 0.5 mL Tris calcium buffer (20 mM Tris, 138 mM NaCl, 2 mM $CaCl_2$, 0.1 % $NaN_3$, pH 8.0). Each washing step consisted of gentle vortexing and centrifugation at $3100 \times g$ for 1 min at 4 °C. The supernatant was discarded, and the pellet was resuspended in 1 mL of freshly prepared 5 mg mL$^{-1}$ Clostridium histolyticum collagenase (Sigma) in Tris calcium buffer. After incubation overnight at 37 °C, the tissue material was centrifuged at $3100 \times g$ for 30 min at 4 °C. The retained pellet was resuspended in 0.5 mL buffer containing 20 mM Tris, 140 mM NaCl, 10 mM ethylenediaminetetraacetic acid, 0.1% $NaN_3$, pH 8.0, and subjected to 10 cycles of homogenization in fresh buffer and centrifugation for 5 min at $3100 \times g$ at 4 °C. The remaining pellet was homogenized in 0.5 mL ice-cold water, centrifuged for 5 min at $3100 \times g$ at 4 °C, and the fibril-containing supernatant was analyzed.

### Protein sequence determination by electrospray-ionization MS

About 2 μg refolded, lyophilized, and glycosylated fibril protein was resuspended and processed by denatured protein gel electrophoresis.

Afterward, the gel band of the fibril protein was cut into 1 mm$^3$ colloidal particles and decolorized by incubating with 1 ml 50% ACN-50% 50 mmol/L $NH_4HCO_3$ solution for 30 min. This operation was repeated until the colloidal particles were colorless. Then, 1 mL 100% ACN 1000 μL was incubated for 30 min, dehydrated, and left to dry at room temperature.

Dried gel slices were reduced with 10 mM dithiothreitol (Sigma-Aldrich), in a water bath at 56 °C for 1 h and subsequently alkylated with 55 mM iodoacetamide (Sigma-Aldrich) for 1 h at 37 °C.

The gel slices were placed in six different protease solutions (0.025 μg/μl trypsin in 50 mM NH4HCO3; 1 μg/μl pepsin in HCl, pH 5.5; 0.05 μg/μl chymotrypsin in 1 mM HCl, 2 μM $CaCl_2$; 0.1 μg/μl Elastase protease in 100 mM Tris-HCl buffer, pH 8.0; Protease K with a substrate mass ratio of 1:40; 0.025 μg/μl Trypsin in 50 mM $NH_4HCO_3$). Each protease was digested overnight at 37 °C.

The resulting peptide is released from the gel section in two steps: the first step is to add 200 μl of a solution containing 5%TFA-50% ACN-45% water (Elastase protease add 100 μl); The second step is to repeat extraction after 1 h in 37 °C water bath, ultrasound for 5 min, centrifugation for 5 min, combined extraction solution, vacuum centrifugation drying, and vacuum drying at 45 °C after desalting.

Samples were separated by liquid chromatography using a Vanquish Core HPLC system (Thermo Fisher Scientific) with an Acclaim PepMap analytical column (150 μm × 150 mm, 3 μm, 100-A aperture, Thermo Fisher Scientific) in combination with a C18 μ-pre-column (PepMap, Thermo Fisher Scientific). A gradient of solvent A [0.1% formic acid] and solvent B [80% ACN, 0.1% formic acid] was used for separation at a flow rate of 600 nL/min. The main column is initially balanced in A mixture containing 5% solvent B and 95% solvent A. For elution, the percentage of solvent B was increased from 5 to 15% in 5 min and followed by an increase from 15 to 40% over 65 min. The fractions of the main column are eluted directly into the ionization module and further analyzed by MS.

Samples were measured using the Orbitrap Fusion Lumos Tribrid Mass Spectrometer (Thermo Fisher Scientific). Primary mass spectrometry obtained a full-scan mass charge ratio ($m/z$) of 150–2000 in Orbitrap with a resolution of 120,000, automatic gain control set to 400,000 ions, and a maximum fill time of 50 ms. Collision-induced dissociation as a crushing method for a single sample set. In a linear ion trap, automatic gain control is set to 5000 ions and a maximum fill time of 118 ms. For MS/MS fragments, fragmentation spectra were detected on an Orbitrap mass spectrometer with a resolution of 60,000.

The secondary mass spectrum was generated by the PEAKS Studio software. The mass spectrometry raw data have been deposited to the ProteomeXchange Consortium via the iProX partner repository under the dataset identifier PXD066587[40,41].

### Negative-staining electron microscopy

Five microliters of an aliquot of fibril sample were applied to a glow-discharged 200 mesh carbon support film (Beijing Zhongjingkeyi Technology Co., Ltd.) for 45 s. Then the grid was washed with 5 μl double-distilled water and followed by another wash of 5 μl 3% w/v uranyl acetate. Another 5 μl 3% w/v uranyl acetate was added to the grid to stain the sample for 45 s. After removing the excess buffer by filter paper, the grid was dried by an infrared lamp. The sample imaging was applied by a Talos L120C microscope (Thermo Fisher) operated at 120 KV.

### Cryo-electron microscopy

An aliquot of 4 μl extracted fibril sample was applied to a glow-discharged holey carbon grid (Quantifoil R1.2/1.3,300 mesh), blotted for 6.5 s, and plunge-frozen in liquid ethane using Thermo Fisher VitrobotMark IV with 95% humidity at 16 °C. The grids were loaded onto a Thermo Fisher Titan Krios microscope operated at 300 kV equipped with a Falcon 4 direct electron detector. EPU software was used for automated data collection according to standard procedures. Magnification at 130,000× was used for imaging, yielding a pixel size of 0.93 Å for the images. The defocus range was set from −0.7 to −1.5 μm. Each micrograph was collected under a dose rate of about 13.24 electrons per Å$^2$ per second, with a total exposure time of 3.92 s, resulting in a total dose of about 52 electrons per Å$^2$. A total of 1626 movies were collected.

### Helical reconstruction

Image stacks were corrected for beam-induced sample motion and implemented dose weighting using MotionCor2[42]. Contrast transfer function estimation of aligned, dose-weighted micrographs was performed by CTFFIND4.1.8[43]. Subsequent image-processing steps, including manual picking, particle extraction, 2D classification, 3D classification, 3D refinement, and postprocessing, were performed using helical reconstruction methods in RELION3.0[44]. In total, 5786 fibrils were picked manually from 1626 micrographs, and 400 pixel boxes were used to extract particles by a 90% overlap scheme. 2D classification of 400-box-size particles was used to calculate the initial twist angle. In regard to helical rise, 4.8 Å was used as the initial value. After several iterations of 2D and 3D classification, particles with the same morphology were picked out. Local searches of symmetry in 3D classification were used to determine the final twist angle and rise value. A featureless cylinder model created by the relion_helix_toolbox program was used as the initial model; 3D classification was performed several times to generate a proper reference map for 3D refinement. 3D refinement of the selected 3D classes with appropriate reference was performed to obtain the final reconstruction.

The final map of the Polymorph A was convergence with the rise of 2.46 Å and the twist angle of −179.58°. Postprocessing was performed to sharpen the map with a B-factor of −46.07 Å$^2$. Based on the gold-standard FSC = 0.143 criteria, the overall resolution was reported as 2.93 Å. The final map of Polymorph B was convergence with the rise of 4.92 Å and the twist angle of 0.87°. Postprocessing was performed to sharpen the map with a B-factor of −35.29 Å$^2$. The overall resolution was reported as 3.00 Å.

## Atomic model building and refinement

Atomic models were subsequently built into the refined maps with COOT[45,46]. The model of the Polymorph A was built based on the model of the cardiac IGLV1-44 fibril (PDB ID: 6IC3). To build the polymorph B model, a single chain from the polymorph A structure was extracted and fitted into the density map of the twist-trimer. For both fibrils, a three-layer model was generated for structure refinement. Structure models were refined by using the real-space refinement program in PHENIX[47].

## Ethical statement

This study was approved by the Ethical Committee of the First Affiliated Hospital, Zhejiang University School of Medicine ([2025B]IIT Ethics Approval No.0278) and was performed in accordance with the Declaration of Helsinki. We have written informed consent from the patient to publish clinical information potentially identifying the individual.

## Reporting summary

Further information on research design is available in the Nature Portfolio Reporting Summary linked to this article.

## Data availability

The reconstructed cryo-EM maps were deposited in the Electron Microscopy Data Bank with the accession codes EMD-66530 (polymorph A) and EMD-64858 (polymorph B). The coordinate files of the fibril models were deposited in the Protein Data Bank with the accession codes 9X45 (polymorph A) and 9V91 (polymorph B). The mass spectrometry raw data were deposited in the ProteomeXchange database under accession number PXD066587. Source data are provided with this paper for Supplementary Fig. 4. Source data are provided with this paper.

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

## Acknowledgements

The authors thank the Cryo-Electron Microscopy Center of the Liangzhu laboratory for the cryo-EM data collection. This project was supported by the National Natural Science Foundation of China Grant (82341206 and 82130003 to H.H.), "Pioneer" and "Leading Goose" R&D Program of Zhejiang (grant number: 2024SSYS0023 to H.H.).

## Author contributions

Conceptualization and supervision by H.H. and K.Z. Investigation and analysis by C.Y., Y.M., H.L., K.L., and K.Z. Funding acquisition and resources by H.H. Review and editing by K.Z. Data visualization by C.Y., Y.M., edited and reviewed by C.Y., Y.M., H.L., K.L., and K.Z. Contribution to and approval of the submitted version by all authors.

## Competing interests

The authors declare no competing interests.
