## [Transparent Peer Review file · Nature Communications]

Cryo-EM structure of renal AL amyloid fibrils from a patient with λ 1 light chain amyloidosis

Corresponding Author: Dr Kun Zhao

Version 0:

Reviewer comments:

Reviewer #1

(Remarks to the Author)

The manuscript by Yu et al. provides a concise and clear structural characterization of amyloid fibrils from renal AL amyloidosis derived from the IGLV1-44 light chain. The authors successfully determine high-resolution cryo-EM structures of two fibril polymorphs and compare them with the previously reported heart-derived IGLV1-44 amyloid structure to extract conclusions about conformational differences.

While the core findings are solid and of interest to the field, I believe the manuscript can be significantly improved in both presentation and clarity. Below, I outline two major concerns and several minor points that should be addressed prior to publication.

Major Comments

1. Symmetry of Polymorph A

The authors state (P4LN145–148) that "Polymorph A displays an approximate two-fold helical symmetry with protofilaments staggered along the helical axis." This is not accurate. The symmetry appears more consistent with a 2_1 helical axis. Furthermore, the sentence "If not considering the pseudo twofold symmetry, the helical twist of the polymorph A fibril would be 0.84° and the helical rise would be 4.92 \AA " again mischaracterizes the symmetry. This is not a "pseudo twofold" but rather a (pseudo) 2_1 screw symmetry. Most concerningly, the density map of Polymorph A visually supports a 2_1 symmetry.

Therefore, the authors should:

- Reprocess the data with 2_1 symmetry and redeposit the maps, or
- Justify in detail why this symmetry was not applied, and convincingly explain their symmetry assignment in the manuscript.

2. Organ-specific conclusions not supported

In several parts of the text (e.g., P1LN34–35 and P7LN240–242), the authors conclude that their results reveal "organ-specific structural variations." This is not supported by the data. The structural differences highlighted in the manuscript derive from sequence variations, not from the tissue of origin. The authors provide no evidence that the organ environment drives differential assembly.

Please revise the abstract and discussion accordingly to reflect that the observed structural differences are sequence-dependent, not organ-specific.

Minor Comments

• Title and Section Headings

The current title ("Cryo-EM structure of amyloid fibrils from renal AL amyloidosis") is too generic and reads more like a review article title. Please revise it to reflect the specific fibril type studied and the key findings. Similarly, the result section titles are generic and should be updated for better scientific clarity, though this is left to the authors' discretion.

• Terminology and Acronyms

- P2LN43: Define "LC" (light chain) here, and remove the definition from P2LN59. Use "LC" throughout afterward.
- P3LN98, 106: "Light chain (LC)" is already defined. Use "LC" only.
- P3LN105, P4LN137–139: "Cryo-electron microscopy (cryo-EM)" was defined in P3LN91. Please omit redundant definitions.
- P4LN127: Use the correct acronym "IGLV," not "IGVL."

• Negative Staining Description

- P4LN137–139: The logical flow is confusing. Authors jump from cryo-EM to negative stain (NS) and back. Please rewrite for clarity: first describe the NS data, then describe cryo-EM.
- P4LN138–139: The NS data are low-resolution and insufficient to distinguish polymorphs, particularly with measurements taken from single fibrils. Please either:

1. Provide statistically robust measurements supporting polymorph discrimination in NS micrographs, or
 2. Revise the text to state only that NS shows fibrils with helical symmetry consistent with amyloid.
- Helical Processing and Figures
 - P4LN140–142: Authors incorrectly state they processed “5,786 fibril segments.” This number refers to filaments. Please correct.
 - The claim that polymorphs are distinguishable in 2D class averages is not supported by the figure. According to Sup Table 1, the separation occurs during 3D classification. Please revise accordingly.
 - P4LN142: Update SI Fig. 3 to include a standard data processing workflow figure, with a representative micrograph, 2D classes, initial model, and final volume. Include all relevant particle numbers and classification steps.
 - P4LN144: In SI Fig. 4, add a horizontal FSC = 0 line for clarity.
 - Structural Description and Labeling
 - P4LN145: In Fig. 1, the twist angle is incorrectly described for polymorph B (0.87°). Please correct.
 - P4LN152–153: Fig. 2a should include a magnified inset of the disulfide bond (Cys22–Cys89). Label Gln16 and Ser95 rather than just “N-” and “C-terminus.”
 - P5LN164–165: Call out Suppl. Fig. S3b,c to better support the statement about structural disorder.
 - P5LN167: Secondary structure annotation (β 1– β 9) is better represented in panel 2c. Please refer to it explicitly.
 - Molecular Interactions
 - P5LN172–173 and P6LN202: Proline cannot participate in π – π interactions. Likely interactions are CH– π or hydrophobic contacts. Please revise.
 - P5LN176: The similarity in β -sheet layering is described qualitatively. Please calculate and report the RMSD of C α atoms for both protofilament chains.
 - P5LN180–181 & P7LN224: Replace “domain” with “region” since these are not independently folded units.
 - P6LN205–207: Include a figure (or additional panels in an existing one) illustrating mutant residues (e.g., Tyr37Phe, Asp86His) and their local environments.
 - P6LN211: The phrase “providing a rationale for clinical.” is incomplete. Clarify whether this refers to treatment, diagnosis, etc.
 - Topology and Comparative Analysis
 - P7LN224–226: The claim about the N→C parallel-to-antiparallel transition requires visual support. Please show this clearly in Fig. 3c/d or a supplementary figure.
 - P7LN234–236: The explanation of the cavity formation by Gln39His and Tyr37Phe is not entirely correct. These residues contribute differently: H39 interacts across protofilaments; F37 flips into the core. Please revise for accuracy and include a structural overlay of the renal and heart fibrils to illustrate the folding difference.
 - P7LN238: Remove “network”—a single salt bridge is not a network.
 - P7LN249: Clarify “Ser25Arg” — is this Ser25 in heart and Arg25 in renal? Please explain.
 - P9LN267–268: The statement that “fibril cores are universally formed by the VL domain” is outdated. See: Schulte T, et al. Nat Commun. 2024 Jul 28;15(1):6359. Update the discussion accordingly to reflect this more nuanced understanding.
 - Figures
 - Fig. 4A: Label clearly which fibril is from kidney and which from heart.
 - Include a structural overlap between the renal and heart IGLV1-44 fibrils to illustrate folding differences.
 - Methods Clarity
 - In “Helical reconstruction,” the initial model generation is unclear. Please specify the software and protocol used to create the initial volume.

While I find the manuscript scientifically sound and the structures relevant to the field, several issues must be addressed—especially regarding the interpretation of symmetry and the overstated conclusion about organ specificity. Once these points are corrected, I believe the manuscript will be suitable for publication in Nature Communications.

Sincerely,
Antonio Chaves-Sanjuan

Reviewer #2

(Remarks to the Author)

The manuscript “Cryo-EM structure of amyloid fibrils from renal AL amyloidosis” by Yu et.al present a structural analysis of fibrils from mutated IGLV 1-44 protein by cryo electron microscopy. Similar studies published previously report high-resolution Cryo-EM structures of immunoglobulin light chain fibrils associated with cardiac amyloidosis, originating from various lambda germline gene segments (Swuec et al., 2019; Radamaker et al., 2019; Radamaker et al., 2021 [two articles]; Schulte et al., 2024).

In the current manuscript, the authors identify and characterize two structurally distinct fibril polymorphs from renal tissue samples, resolved at up to 3.0 Å resolution. They describe the chemical interactions that stabilize the fibril cores and analyze the surface charge distributions. Additionally, they highlight key structural differences between IGLV 1-44 derived fibrils in renal versus cardiac amyloidosis. Based on these differences, the authors propose potential mechanisms underlying organ tropism and tissue-specific pathogenic pathways of fibril deposition.

While the study contributes valuable structural insights, certain aspects of the manuscript could be improved to better distinguish it from existing literature and ensure it does not appear incremental relative to prior Cryo-EM studies on fibrils from systemic amyloidosis.

1. The manuscript appropriately focuses on the structural analysis of fibrils using Cryo-EM, the study also touches on organ-specific deposition patterns and pathogenic mechanisms. These aspects, however, are addressed only superficially and

lack clear integration with the structural findings. The authors should explicitly state the primary goals of the study at the outset and clarify how the structural data are intended to support these conclusions.

2. Line 203-205 – It is unclear what the aggregation kinetics are being compared against. Are the authors referring to aggregation kinetics derived from cardiac derived IGLV 1-44 fibrils?

3. Line 210-211 – Detecting mutation profiles.....: this statement needs further elaboration. How could specific mutations influence therapeutic choices?

4. Given the significance of in vitro systems in studying aggregation kinetics and therapeutic screening, it would be valuable to compare the structure of in vitro fibrils (seeded with renal tissue-derived fibrils) to the ex vivo structures presented. Such a comparison could further enhance the understanding of in vitro models in replicating disease-relevant fibril structures.

5. Lines 237-242 – The discussion on changes in residues involved in the formation of intermolecular salt bridges leading to organ-specific deposition patterns is intriguing but are these changes sufficient to explain the observed tropism?

6. The role of surface charge and hydrophobicity in fibril formation and stability must be elaborated. How do these physicochemical properties in renal IGLV 1-44 fibrils compare with those found in cardiac-derived fibrils?

7. What is structure shown in figure 3d? The PDB:6IC3 mentioned belongs to cardiac IGLV 1-44 fibrils and not natively folded LC variable domain. It is difficult to ascertain the N , C -terminus orientation of the native LC from fig 3d. The figure should be revised for accuracy, and the legend should be corrected accordingly.

8. What is the key step in the misfolding of the natively folded light chain proteins that initiate fibril formation and arrangements into polymorph A and B?

Minor points: Check the manuscript for typographical errors – especially in the references.

Version 1:

Reviewer comments:

Reviewer #1

(Remarks to the Author)

The authors have done an excellent job addressing all the points raised during the previous review round. The manuscript has improved substantially in clarity, rigor, and overall presentation. The experimental design and data interpretation are now well explained, and the figures and supplementary materials are clear and consistent with the conclusions.

I particularly appreciate the care and thoroughness with which the revisions were carried out — the responses are detailed and convincing, and the resulting manuscript is stronger and more readable.

Congratulations on the excellent revision. I have no further concerns and recommend the paper for publication in its current form.

Antonio Chaves-Sanjuan

Reviewer #2

(Remarks to the Author)

I am satisfied with the revisions the authors have made in response to my review comments. The manuscript has been adequately improved, and no further modifications are required from my side.

Point-by-point response to the reviewers' comments:

* The reviewer's comments are in *Italic*. Author's responses are in blue. All revisions are highlighted in yellow in the revised manuscripts.

Reviewers' comments:

Reviewer #1:

The manuscript by Yu et al. provides a concise and clear structural characterization of amyloid fibrils from renal AL amyloidosis derived from the IGLV1-44 light chain. The authors successfully determine high-resolution cryo-EM structures of two fibril polymorphs and compare them with the previously reported heart-derived IGLV1-44 amyloid structure to extract conclusions about conformational differences.

While the core findings are solid and of interest to the field, I believe the manuscript can be significantly improved in both presentation and clarity. Below, I outline two major concerns and several minor points that should be addressed prior to publication.

REPLY: We sincerely thank this reviewer for the positive assessment of this work. We especially thank the reviewer for his/her insightful suggestions provided below, which are of great help to improve this manuscript.

Major Comments:

1. Symmetry of Polymorph A

The authors state (P4LN145–148) that "Polymorph A displays an approximate two-fold helical symmetry with protofilaments staggered along the helical axis." This is not accurate. The symmetry appears more consistent with a 2_1 helical axis.

Furthermore, the sentence "If not considering the pseudo twofold symmetry, the helical twist of the polymorph A fibril would be 0.84° and the helical rise would be 4.92 \AA " again mischaracterizes the symmetry. This is not a "pseudo twofold" but rather a (pseudo) 2_1 screw symmetry. Most concerningly, the density map of Polymorph A visually supports a 2_1 symmetry.

Therefore, the authors should:

- *Reprocess the data with 2_1 symmetry and redeposit the maps, or*
- *Justify in detail why this symmetry was not applied, and convincingly explain their symmetry assignment in the manuscript.*

REPLY: We thank the reviewer for this insightful and critical comment regarding the symmetry of the Polymorph A. The reviewer is absolutely correct. We apologize for the inaccuracies in our initial symmetry description.

Following the reviewer's recommendation, we have taken the following actions:

1. Data Reprocessing: We have accordingly reprocessed the cryo-EM data for Polymorph A by imposing a 2_1 helical symmetry during the final 3D reconstruction. This has resulted in a significantly improved density map with better-defined features.
2. Map and Model Redeposition: The newly reconstructed map and the refined atomic model, which now strictly adheres to the 2_1 symmetry, have been redeposited in the respective public databases. The updated accession codes (or PDB/EMDB IDs) are now provided in the revised manuscript.
3. Manuscript Revision: The text has been revised throughout to correctly describe the 2_1 screw symmetry, and all relevant figures and parameters have been updated.

We believe that the application of the correct symmetry, as suggested by the reviewer, has strengthened the quality and accuracy of our structural model and the overall manuscript. We are grateful for the reviewer's expertise, which has been invaluable in correcting this important aspect of our work.

“.....Consistent with the density map, polymorph A fibril displays a (pseudo) 2_1 screw symmetry, characterized by a helical twist of -179.58° and a rise of 2.46 \AA”

“.....we resolved contain two protofilaments: polymorph A fibril displays a (pseudo) 2_1 screw symmetry, while polymorph B displays $C1$ helical symmetry.....”

2. Organ-specific conclusions not supported

In several parts of the text (e.g., P1LN34–35 and P7LN240–242), the authors conclude that their results reveal “organ-specific structural variations.” This is not supported by the data. The structural differences highlighted in the manuscript derive from sequence variations, not from the tissue of origin. The authors provide no evidence that the organ environment drives differential assembly.

Please revise the abstract and discussion accordingly to reflect that the observed structural differences are sequence-dependent, not organ-specific.

REPLY: We thank the reviewer for this critical comment. We agree that the term “organ-specific structural variations” was inaccurate and could be misleading, as it might imply that the organ environment directly drives the assembly of different fibril structures. This was not our intention.

As the reviewer correctly pointed out, our data demonstrate that the structural differences originate from sequence variations (i.e., the mutations). Our intended meaning was that these sequence-dependent structural variations could, in turn, influence the organ tropism (i.e., the propensity to deposit in specific organs) of the amyloid fibrils.

To accurately reflect this, we have revised the text to use “organ-tropism” and related phrases in the abstract, introduction and discussion.

“.....Despite significant advances in understanding amyloid fibril structure in AL amyloidosis, the molecular mechanism of organ-tropism in fibril deposition remains unclear.....”

“.....These clinical challenges underscore the urgent need for advances in understanding the molecular pathogenesis of the disease and the mechanisms underlying organ-tropism amyloid deposition.”

“.....These findings reveal pronounced molecular assembly differences between AL amyloid fibrils derived from distinct tissues (Fig. 4c), establishing a structural basis for how mutation-driven fibril polymorphism contributes to organ tropism.”

Minor Comments:

1. Title and Section Headings

The current title ("Cryo-EM structure of amyloid fibrils from renal AL amyloidosis") is too generic and reads more like a review article title. Please revise it to reflect the specific fibril type studied and the key findings. Similarly, the result section titles are generic and should be updated for better scientific clarity, though this is left to the authors' discretion.

REPLY: We are grateful to the reviewer for their constructive feedback, which has helped us improve the clarity and impact of our manuscript.

Regarding the manuscript title:

We completely agree that the previous title was too generic. As suggested, we have revised it to specifically highlight the key fibril type and our major finding. The new title is:

“Cryo-EM structure of renal AL amyloid fibrils from a patient with λ 1 light chain amyloidosis”

We believe this new title is more precise, informative, and better reflects the specific structural insights presented in our study.

Regarding the result section titles:

We are grateful to the reviewer for this valuable feedback. We agree that the original section titles were too generic. Accordingly, we have updated them to precisely articulate the scientific advances in each section. We are confident that the new titles provide a much clearer and more compelling guide for the reader.

- 1、 Extraction and sequence analysis of amyloid fibrils from a patient renal biopsy*
- 2、 Cryo-EM structural determination of the renal AL fibril*
- 3、 Salt bridges and hydrophobic interactions mediate stability in the fibril protein fold*
- 4、 Implication of patient-specific mutations for fibril morphology*
- 5、 Comparative analysis with cardiac fibrils and native light chain structures”*

We hope that our revised title, combined with the enhanced clarity in the result narratives, fully addresses the reviewer's concern and improves the overall scientific clarity of the manuscript.

2. Terminology and Acronyms

- 1) P2LN43: Define "LC" (light chain) here, and remove the definition from P2LN59. Use "LC" throughout afterward.
- 2) P3LN98, 106: "Light chain (LC)" is already defined. Use "LC" only.
- 3) P3LN105, P4LN137–139: "Cryo-electron microscopy (cryo-EM)" was defined in P3LN91. Please omit redundant definitions.
- 4) P4LN127: Use the correct acronym "IGLV," not "IGVL."

REPLY: We apologize for the mistake. They have been corrected in the revised manuscript. Proofreading has also been applied.

1) We defined "LC" (light chain) here at the very beginning, and "LC" was directly used in the subsequent content.

“Systemic light-chain (AL) amyloidosis is a disorder characterized by the abnormal aggregation of immunoglobulin light chains (LCs) into oligomers and amyloid fibrils.....”

“.....As a result, the LC sequences derived from either the λ or κ loci are nearly unique to each individual patient.....”

2) We modified the original text and directly used LC.

“.....highlighting the immunoglobulin LC sequence as a key determinant of fibril structure.....”

“.....we employed cryo-EM to determine the high-resolution three-dimensional structure of LC amyloid fibrils extracted from the kidney tissue of a patient with AL amyloidosis.....”

3) We have corrected the errors in the original manuscript.

“.....we employed cryo-EM to determine the high-resolution three-dimensional structure of LC amyloid fibrils extracted from the kidney tissue of a patient with AL amyloidosis.....”

“.....To determine the atomic structures, we subsequently employed cryo-EM.....”

4) We have corrected the errors in the original manuscript.

“.....Previous studies have demonstrated that the immunoglobulin light chain variable region (IGLV) is associated with organ tropism.....”

3. Negative Staining Description

1) *P4LN137–139: The logical flow is confusing. Authors jump from cryo-EM to negative stain (NS) and back. Please rewrite for clarity: first describe the NS data, then describe cryo-EM.*

REPLY: We thank the reviewer for pointing out the issue with the logical flow. We have rewritten the section as suggested, now presenting the NS data first to establish the initial morphological characterization, followed by the high-resolution structural insights from cryo-EM. This restructuring creates a more coherent narrative from initial screening to high-resolution structure determination. We have revised the manuscript accordingly.

“To investigate the structure of the extracted fibrils, we first performed negative-staining TEM. This initial analysis revealed the presence of fibrils with the characteristic helical symmetry of amyloid but was insufficient to resolve high-resolution details (SI Fig. 2). To determine the atomic structures, we subsequently employed cryo-EM. Following cryo-EM data collection, we used RELION for helical reconstruction of this fibril species. Three-dimensional (3D) classification from helical image processing of 5,786 filaments revealed two major polymorphs of twisted fibrils, herein designated as polymorphs A and B (SI Fig. 3)”

2) P4LN138–139: *The NS data are low-resolution and insufficient to distinguish polymorphs, particularly with measurements taken from single fibrils.*

Please either:

- 1. Provide statistically robust measurements supporting polymorph discrimination in NS micrographs, or*
- 2. Revise the text to state only that NS shows fibrils with helical symmetry consistent with amyloid.*

REPLY: We sincerely thank the reviewer for this critical observation. We agree that the previous narrative flow was suboptimal and could cause confusion. As suggested, we have entirely restructured the Results section to present a linear and logical progression of our experimental workflow. We now first describe the NS-TEM data (which served as an initial quality check and morphological assessment), followed by the detailed presentation of the cryo-EM data (which provided the high-resolution information for structural analysis and polymorph distinction). Meanwhile, we have also made corresponding modifications to SI fig2. We believe this revision significantly improves the clarity and readability of the manuscript.

“To investigate the structure of the extracted fibrils, we first performed negative-staining TEM. This initial analysis revealed the presence of fibrils with the characteristic helical symmetry of amyloid but was insufficient to resolve high-resolution details (SI Fig. 2). To determine the atomic structures, we subsequently employed cryo-EM. Following cryo-EM data collection, we used RELION for helical reconstruction of this fibril species. Three-dimensional (3D) classification from helical image processing of 5,786 filaments revealed two major polymorphs of twisted fibrils, herein designated as polymorphs A and B (SI Fig. 3).”

Iglv1-44 fibril

Figure S2 Negative-staining TEM images reveal the characteristic helical symmetry of IGLV1-44 fibrils.

4. Helical Processing and Figures

1) *P4LN140–142: Authors incorrectly state they processed “5,786 fibril segments.” This number refers to filaments. Please correct.*

REPLY: We apologize for the mistake. It has been corrected in the revised manuscript. Proofreading has also been applied.

“Two-dimensional (2D) and three-dimensional (3D) classification from helical image processing of 5,786 filaments revealed two major polymorphs of twisted fibrils upon 3D analysis (SI Fig. 3).”

2) *The claim that polymorphs are distinguishable in 2D class averages is not supported by the figure. According to Sup Table 1, the separation occurs during 3D classification. Please revise accordingly.*

REPLY: We thank the reviewer for this critical insight and for highlighting the need for greater precision in our data interpretation. We fully agree that the distinction between the two polymorphs is conclusively demonstrated by the 3D classification results, as correctly noted in Supplementary Table 1, and not by 2D class averages alone. As suggested, we have revised the text on page 4 to accurately reflect the source of this conclusion. The sentence now clearly states that the two polymorphs were identified through “upon 3D analysis”. We apologize for the confusion.

“.....Two-dimensional (2D) and three-dimensional (3D) classification from helical image processing of 5,786 filaments revealed two major polymorphs of twisted fibrils upon 3D analysis (SI Fig. 3).”

3) P4LN142: Update SI Fig. 3 to include a standard data processing workflow figure, with a representative micrograph, 2D classes, initial model, and final volume. Include all relevant particle numbers and classification steps.

REPLY: We thank the reviewer for this excellent suggestion. We have now updated SI Fig. 3 as recommended. The new figure provides a comprehensive workflow, including: (i) a representative raw micrograph, (ii) representative 2D class averages, (iii) the initial model, and (iv) the final reconstructed volume. We have also included a detailed flowchart that outlines the data processing steps, along with the corresponding particle numbers at each classification stage. This revision offers a clearer and more complete overview of our data processing strategy.

Figure S3 Work flow of steps of the reconstruction process. A representative micrograph and 2D class average are shown at the top left and top right, respectively. Scale bars are shown

in the pictures. Data statistics, no. of micrographs, and no. of particles used in image processing are listed. Based on an initial reconstruction, 3D classification separated two subsets exhibiting different density paths. Through several further classification and refinement steps, the final reconstructions were obtained.

4) P4LN144: In SI Fig. 4, add a horizontal FSC = 0 line for clarity.

REPLY: Following the reviewer's suggestion, we added the horizontal FSC = 0 line for clarity as shown in SI Fig. 4.

Figure S4 FSC curve of the reconstructed density

5. Structural Description and Labeling

1) P4LN145: In Fig. 1, the twist angle is incorrectly described for polymorph B (0.87°). Please correct.

REPLY: We sincerely apologize for the mistake. We have now corrected Figure 1 with the accurate value of 0.87° for polymorph B.

2) P4LN152–153: Fig. 2a should include a magnified inset of the disulfide bond (Cys22–Cys89). Label Gln16 and Ser95 rather than just “N-” and “C-terminus.”

REPLY: We thank the reviewer for their insightful suggestion, which has significantly improved the clarity and informational value of Figure 2. As suggested, we have included a new inset panel that provides a clear, close-up view of the disulfide bond between Cys22 and Cys89. Additionally, the generic “N-terminus” and “C-terminus” labels have been replaced with the specific residue names “Gln16” and “Ser95”, respectively, to provide more precise structural annotation.

3) P5LN164–165: Call out Suppl. Fig. S3b,c to better support the statement about structural disorder.

REPLY: We are grateful to the reviewer for this comment. In response, and following our update to the workflow in Figure S3, we have revised the text to cite this updated figure. The revised figure provides direct visual support for the described structural disorder.

“.....The N- and C-termini of the protein were surrounded by diffuse density (SI Fig. 3)”

4) P5LN167: Secondary structure annotation ($\beta 1$ – $\beta 9$) is better represented in panel 2c. Please refer to it explicitly.

REPLY: We thank the reviewer for this insightful suggestion. We have now explicitly referenced Panel 2c in the relevant section of the Results part to ensure clarity and direct readers to this improved visualization.

“.....we designated as $\beta 1$ through $\beta 9$, respectively (Fig. 2c, d)”

6. Molecular Interactions

1) P5LN172–173 and P6LN202: Proline cannot participate in π - π interactions. Likely interactions are CH- π or hydrophobic contacts. Please revise.

REPLY: We sincerely thank the reviewer for catching this important inaccuracy and for providing the correct explanation. We fully agree with the reviewer that proline is not capable of engaging in π - π stacking interactions. We have revised the manuscript accordingly to remove all incorrect references to π - π stacking involving Pro41.

“.....Polymorph A’s interface is stabilized by two pairs of inter-chain aromatic residue interactions, specifically a symmetric CH- π stacking between His39-Pro41 and Pro41-His39, markedly enhancing interface stability (Fig. 2a). In contrast, polymorph B maintains protofilament interactions through a single CH- π interaction between His39 and Pro41, with an interface topology exhibiting asymmetric features.....”

“.....enabling a cross-chain CH- π stacking interaction with the adjacent Pro41 residue of a neighboring protofilament subunit.....”

2) P5LN176: The similarity in β -sheet layering is described qualitatively. Please calculate and report the RMSD of C α atoms for both protofilament chains.

REPLY: We thank the reviewer for this excellent suggestion. We have now calculated the C α RMSD between the two protofilaments to quantitatively assess their structural similarity. Following structural superposition, the RMSD for C α atoms between the two protofilaments is 0.001 Å. This very low value provides strong quantitative support for our original statement that the core structures of the protofilaments are virtually identical. We have revised the manuscript to include this new data.

“.....Notably, both polymorphs display an identical β -sheet layering pattern within the core of individual protofilaments, which is underscored by a very low C α RMSD of 0.001 Å between the two protofilaments.....”

3) P5LN180–181 & P7LN224: Replace “domain” with “region” since these are not independently folded units.

REPLY: We apologize for the confusion. We agree entirely that the term "region" is more accurate than "domain" for describing these segments, as they are not independently folded structural units. We have carefully gone through the manuscript and replaced all instances where "domain" was used inaccurately with the more appropriate term "region".

“.....The N-terminal region is dominated by positive electrostatic potential (blue), whereas the C-terminal region exhibits a negative potential (red) (Fig. 3a)”

“.....which covalently links the N- and C-terminal regions of the protein”

4) P6LN205–207: Include a figure (or additional panels in an existing one) illustrating mutant residues (e.g., Tyr37Phe, Asp86His) and their local environments.

REPLY: We thank the reviewer for the comment. The local environments of the Tyr37Phe and Asp86His mutant residues, including their salt-bridge networks, are already illustrated in Figure 3b. We have now inserted a reference to Figure 3b in the relevant section of the text to more prominently highlight these specific residues and their interactions, thereby enhancing clarity. Meanwhile, we have also added magnified local environment maps of the mutant residues (Tyr37Phe, Asp86His) in fig 3b to more clearly display various interactions.

“.....The Tyr37Phe mutation (Fig. 3b) may strengthen the van der Waals forces and hydrophobic synergism between β -sheet layers of the fibril, further stabilizing the fibril. Additionally, the Asp86His mutation (Fig. 3b) can form a salt bridge with Glu84, reinforcing the internal fibril architecture”

5) P6LN211: The phrase “providing a rationale for clinical.” is incomplete. Clarify whether this refers to treatment, diagnosis, etc.

REPLY: We apologize for the confusion. We have revised the sentence to complete the thought. The sentence now clearly states that this approach provides a rationale for first-line treatment options in clinical practice.

“.....Detecting mutation profiles in patient light chains could therefore guide the selection of targeted therapies, providing a rationale for first-line treatment options in clinical practice. For instance, if a mutation profile closely aligns with that characteristic of cardiac-derived fibrils, a cardiotropic treatment strategy would be prioritized. Conversely, the detection of a mutation profile associated with renal amyloidosis would justify a renally targeted approach. However, it is important to note that the existing maps of organ-specific mutational signatures remain incomplete. This evolving framework for mutational profiling nonetheless allows for a more personalized and biologically rational selection of first-line therapies, ultimately aiming to improve treatment outcomes.”

7. Topology and Comparative Analysis

1) P7LN224–226: The claim about the N→C parallel-to-antiparallel transition requires visual support. Please show this clearly in Fig. 3c/d or a supplementary figure.

REPLY: We thank the reviewer for this important suggestion. We fully agree that providing direct visual support for the key finding of the “N→C parallel-to-antiparallel transition” is crucial for readers to appreciate the extent of the conformational rearrangement. As recommended, we have created a new Supplementary Figure (SI Fig. 6), which provides a side-by-side structural comparison of the natively folded IGLV1-44 and the renal IGLV1-44 fibril. We are confident that this new supplementary figure provides direct structural evidence for our claim of a “parallel-to-antiparallel transition,” making it immediately clear and incontrovertible. We have now referenced this figure in the main text and discuss this finding. We thank you again for this valuable comment, which has significantly improved the clarity and impact of our manuscript.

2) P7LN234–236: *The explanation of the cavity formation by Gln39His and Tyr37Phe is not entirely correct. These residues contribute differently: H39 interacts across protofilaments; F37 flips into the core. Please revise for accuracy and include a structural overlay of the renal and heart fibrils to illustrate the folding difference.*

REPLY: We thank the reviewer for this insightful comment and for highlighting the need for a more precise mechanistic description. The reviewer is absolutely correct. We have now revised the text to accurately describe the distinct roles of these two residues: the Gln39His mutation primarily facilitates inter-protofilament interactions, while the Tyr37Phe mutation allows the residue to flip into the protein core. We now clearly state that the combined effect of these two distinct mechanisms is the formation of the enlarged cavity.

“.....Further structural analysis delineates the mechanisms by which the Gln39His and Tyr37Phe mutations reshape the fibril core: Gln39His mediates inter-protofilament contacts, while Tyr37Phe causes a inward flipping of the residue. This dual rearrangement culminates in the formation of a larger internal cavity specific to the renal fibril.”

Additionally, as requested, we have included a new structural overlay comparing the renal and heart fibrils to visually underscore this folding difference. This new panel has been added as Figure 4c. We believe these revisions have significantly improved the accuracy and clarity of our structural description.

3) P7LN238: Remove “network”—a single salt bridge is not a network.

REPLY: We thank the reviewer for this precise correction. We agree that the term “network” is inaccurate for describing a single salt bridge. We have therefore removed the word “network” throughout the manuscript to improve scientific accuracy.

“.....forms a stable intermolecular salt bridge with the neighboring Glu84 residue.....”

4) P7LN249: Clarify “Ser25Arg” — is this Ser25 in heart and Arg25 in renal? Please explain.

REPLY: We thank the reviewer for raising this point, which allows us to clarify an important detail. The nomenclature “Ser25Arg” does not refer to a difference between tissues (e.g., heart vs. renal). Instead, it indicates a somatic mutation from serine (Ser) to arginine (Arg) at position 25 when comparing the amyloid fibril sequence (derived from the heart in this case) to the IGLV1-44 germ line sequence.

To prevent any confusion for the reader, we have revised the manuscript to explicitly state this.

“.....whereas in the cardiac fibril, structural stabilization is mediated by a salt bridge interaction between Asp86 and Arg25.....”

5) P9LN267–268: *The statement that “fibril cores are universally formed by the VL domain” is outdated. See:*

Schulte T, et al. Nat Commun. 2024 Jul 28;15(1):6359.

Update the discussion accordingly to reflect this more nuanced understanding.

REPLY: We thank the reviewer for this insightful comment. We agree that the statement “fibril cores are universally formed by the VL domain of precursor λ -light chains” is overly absolute and does not reflect recent structural advances. As rightly pointed out by the reviewer, the study by Schulte et al. (2024, Nat Commun) presents the first cryo-EM structure of an AL amyloid fibril (AL59) that incorporates an extended fragment of the constant domain (CL) within its core. We thank the reviewer for bringing this important point to our attention, which has helped us improve the manuscript.

We have modified the sentence to reflect this more nuanced understanding. The revised text now reads:

“.....fibril cores are typically formed by the VL domain of precursor λ -light chains, although emerging evidence demonstrates that the constant domain (CL) can also be incorporated into the core structure in some cases”

8. Figures

1) *Fig. 4A: Label clearly which fibril is from kidney and which from heart.*

REPLY: Thank you for this important comment. We agree that clearly labeling the tissue origin of the fibrils is essential for the reader's understanding of the experimental results. As suggested, we have revised Figure 4a and 4b. The panels now include clear text labels directly on the images to unambiguously indicate the origin of each fibril or structural model. This modification ensures that readers can easily distinguish between the structures of the fibrils from different tissue origins, avoiding any potential confusion. We thank you again for helping us improve the clarity and precision of our manuscript.

2) Include a structural overlap between the renal and heart IGLV1-44 fibrils to illustrate folding differences.

REPLY: We thank the reviewer for this valuable comment. We agree that a structural overlap between the renal and heart IGLV1-44 fibrils is crucial to illustrate their folding differences, as this provides a direct visual and quantitative comparison of their atomic-level structures. As suggested, we have modified the manuscript accordingly. Specifically, we have added a new panel (Figure 4c) to Figure 4, which presents a structural alignment of the renal (colored in light pink) and heart (colored in pale yellow) IGLV1-44 fibrils. This structural overlap clearly demonstrates that while the two fibrils share some basic architectural features, there are significant conformational differences, particularly in key loop regions.

9. Methods Clarity

1) In “Helical reconstruction,” the initial model generation is unclear. Please specify the software and protocol used to create the initial volume.

REPLY: We sincerely thank the reviewer for pointing out this critical omission and the lack of clarity in our original description. The reviewer is absolutely correct. We apologize for the error in our initial manuscript, which incorrectly stated that “The 3D initial model was built by selected 2D classes”. This was a mistake in the method description. To address this comment, we have now corrected the manuscript as follows:

The initial model for helical reconstruction was, in fact, a featureless cylinder generated de novo using the `relion_helix_toolbox` program (a part of the RELION software suite), not from selected 2D classes. This is a standard and robust approach for ab initio helical reconstruction to avoid potential reference bias.

We have now revised the Methods section to precisely state: “An initial featureless cylinder model was created using the `relion_helix_toolbox` program within RELION-3.1 and used as the reference for subsequent 3D classification and refinement.”

Furthermore, to enhance clarity and transparency, we have also updated the supplementary Figure S3 (the workflow figure) to explicitly include a visualization of this featureless cylinder initial model in the diagram, as suggested by the reviewer's

comment. This provides a clear, visual representation of the starting point for our helical processing pipeline.

We believe these corrections have now accurately and clearly specified the software and protocol used for initial model generation, and we thank the reviewer again for ensuring the methodological rigor of our work.

“.....A featureless cylinder model created by the `reliion_helix_toolbox` program was used as initial model; 3D classification was performed several times to generate a proper reference map for 3D refinement.....”

Reviewer #2 (Remarks to the Author):

The manuscript “Cryo-EM structure of amyloid fibrils from renal AL amyloidosis” by Yu et.al present a structural analysis of fibrils from mutated IGLV 1-44 protein by cryo electron microscopy. Similar studies published previously report high-resolution Cryo-EM structures of immunoglobulin light chain fibrils associated with cardiac amyloidosis, originating from various lambda germline gene segments (Swuec et al., 2019; Rademaker et al., 2019; Rademaker et al., 2021 [two articles]; Schulte et al., 2024).

In the current manuscript, the authors identify and characterize two structurally distinct fibril polymorphs from renal tissue samples, resolved at up to 3.0 Å resolution. They describe the chemical interactions that stabilize the fibril cores and analyze the surface charge distributions. Additionally, they highlight key structural differences between IGLV 1-44 derived fibrils in renal versus cardiac amyloidosis. Based on these differences, the authors propose potential mechanisms underlying organ tropism and tissue-specific pathogenic pathways of fibril deposition.

While the study contributes valuable structural insights, certain aspects of the manuscript could be improved to better distinguish it from existing literature and ensure it does not appear incremental relative to prior Cryo-EM studies on fibrils from systemic amyloidosis.

REPLY: We are grateful to the reviewer for their thoughtful assessment and constructive comments, which have helped strengthen our manuscript. We also appreciate the chance to elaborate on how our study provides novel and distinct contributions to the field.

1. The manuscript appropriately focuses on the structural analysis of fibrils using Cryo-EM, the study also touches on organ-specific deposition patterns and pathogenic mechanisms. These aspects, however, are addressed only superficially and lack clear integration with the structural findings. The authors should explicitly state the primary goals of the study at the outset and clarify how the structural data are intended to support these conclusions.

REPLY: We thank the reviewer for this insightful comment. We agree that the term "organ-specific" was inaccurate in this context. Our intention was to convey that the mutation alters the LC sequence, which in turn leads to distinct amyloid fibril structures in the heart and kidneys. These structural differences ultimately contribute to differing organ tropism of the fibrils, resulting in their deposition in different organs. We sincerely appreciate the reviewer's feedback and have revised the terms "organ-

specific" to "organ tropism" in both the Abstract and Discussion sections to better reflect our intended meaning.

“.....Despite significant advances in understanding amyloid fibril structure in AL amyloidosis, the molecular mechanism of organ-tropism in fibril deposition remains unclear.....”

“.....These findings reveal pronounced molecular assembly differences between AL amyloid fibrils derived from distinct tissues (Fig. 4c), establishing a structural basis for how mutation-driven fibril polymorphism contributes to organ tropism.”

2. Line 203-205 – It is unclear what the aggregation kinetics are being compared against. Are the authors referring to aggregation kinetics derived from cardiac derived IGLV 1-44 fibrils?

REPLY: We thank the reviewer for this question, which allows us to clarify an important point. The comparison of aggregation kinetics is made against the native, non-amyloidogenic state of the protein subunits themselves, not against fibrils derived from other organs. This benchmark is used to demonstrate how the identified interaction enhances the propensity for aggregation relative to the protein's normal state. We apologize for the lack of clarity in the original manuscript and have revised the text to explicitly state "...compared to their native state" to prevent any future misunderstanding.

“.....This interaction facilitates the formation of a unique fibril interface, leading to a dimer architecture distinct from other amyloid fibrils. It thereby constitutes a core structural basis for IGLVI-44 fibril assembly and enhances the aggregation kinetics of the protein subunits compared to their native or non-amyloidogenic states”

3. Line 210-211 – Detecting mutation profiles.....: this statement needs further elaboration. How could specific mutations influence therapeutic choices?

REPLY: We thank the reviewer for raising this important point. We agree that the connection between mutations and therapy should be made more explicit. We have now elaborated on this in the manuscript. The revised text now explains how mutations that dictate fibril structure could influence the efficacy of amyloid-depleting therapies and aid in personalizing treatment strategies.

“.....Detecting mutation profiles in patient light chains could therefore guide the selection of targeted therapies, providing a rationale for first-line treatment options in clinical practice. For instance, if a mutation profile closely aligns with that characteristic of cardiac-derived fibrils, a cardiotropic treatment strategy would be prioritized. Conversely, the detection of a mutation profile associated with renal amyloidosis would justify a renally targeted

approach. However, it is important to note that the existing maps of organ-specific mutational signatures remain incomplete. This evolving framework for mutational profiling nonetheless allows for a more personalized and biologically rational selection of first-line therapies, ultimately aiming to improve treatment outcomes.”

4. Given the significance of in vitro systems in studying aggregation kinetics and therapeutic screening, it would be valuable to compare the structure of in vitro fibrils (seeded with renal tissue-derived fibrils) to the ex vivo structures presented. Such a comparison could further enhance the understanding of in vitro models in replicating disease-relevant fibril structures.

REPLY: We thank the reviewer for their insightful suggestion. We fully agree that a direct comparison between in vitro-generated fibrils and those derived from renal tissues would strengthen the relevance of our in vitro model.

In our in vitro experiments, we employed kidney tissue-derived seeds to induce fibril formation under various conditions such as different pH levels, salt concentrations, and temperatures. Although this approach yielded a diversity of fibrillar morphologies, these structures differed from patient-derived fibrils, suggesting that chemical modifications within the physiological microenvironment play a crucial role in shaping fibril architecture.

After multiple attempts, these in vitro fibrils exhibited heterogeneous morphology under negative-stain EM imaging, as shown in the figure below. Our negative-stain EM data revealed that the in vitro fibrils are structurally heterogeneous, with poorly defined helical symmetry, pronounced twisting, and a tendency to aggregate. These properties hindered homogeneous particle alignment and prevented high-resolution structure determination by cryo-EM. This suggests that, unlike their in vivo counterparts which benefit from tissue-specific microenvironments, fibrils formed in vitro may lack the structural homogeneity required for such analysis. We thank the reviewer again for emphasizing this valuable point.

5. Lines 237-242 – *The discussion on changes in residues involved in the formation of intermolecular salt bridges leading to organ-specific deposition patterns is intriguing but are these changes sufficient to explain the observed tropism?*

REPLY: We agree with the reviewer's insightful comment that organ tropism is undoubtedly multifactorial. Our data demonstrate that these changes are a powerful and previously delineated structural feature that can predispose fibrils to deposit in specific organs by altering their surface electrostatic potential. While we do not claim this to be the exclusive factor, we posit it as a fundamental one that establishes a structural basis for electrostatic interactions with the tissue microenvironment. Future work will be essential to integrate our findings with other potential determinants to build a comprehensive model.

6. *The role of surface charge and hydrophobicity in fibril formation and stability must be elaborated. How do these physicochemical properties in renal IGLV 1-44 fibrils compare with those found in cardiac-derived fibrils?*

REPLY: We thank the reviewer for this excellent and critical point, which allows us to deepen the mechanistic discussion of our findings. As suggested, we have now extensively elaborated on the roles of surface charge and hydrophobicity in a paragraph within the results section.

Specifically, we have added a new Supplementary Figure 5 (SI Fig. 5) comparing the surface charge distribution and hydrophobicity of a cardiac-derived AL fibril with our renal fibril. The analysis reveals that the renal IGLV1-44 fibril exhibits a richer and more complex surface charge distribution relative to its cardiac counterpart, a feature we propose as an adaptation to the high-ionicity microenvironment of the kidney, potentially influencing fibril stability and assembly.

“.....The fibril surface exhibits a pronounced charge distribution pattern, with a molecular surface highly enriched in charged and polar amino acid residues. Notably, compared to cardiac fibrils (SI Fig. 5a, b), our kidney-derived fibrils display a more abundant and complex surface charge pattern, which is likely an adaptation to the charge-rich microenvironment of the kidney.....”

7. What is structure shown in figure 3d? The PDB:6IC3 mentioned belongs to cardiac IGLV 1-44 fibrils and not natively folded LC variable domain. It is difficult to ascertain the N, C -terminus orientation of the native LC from fig 3d. The figure should be revised for accuracy, and the legend should be corrected accordingly.

REPLY: We sincerely thank the reviewer for this insightful comment and for pointing out the inaccuracy in our original labeling of Figure 3D. We acknowledge the error in referencing PDB:6IC3, which indeed corresponds to cardiac IGLV 1-44 fibrils rather than the natively folded LC variable domain. We have now corrected the figure legend to accurately describe the structure shown in Figure 3D as the natively folded IGLV1-44.

“Fig. 3 Surface properties and mutations of the IGLV1-44 fibril. (d) Location of mutations in the corresponding, natively folded LC VL domains that are based on the GL segments IGLV1-44*01(PDB entry 1BJM). Mutations are colored in green.”

Regarding the orientation of the N and C termini, we agree with the reviewer that this was not clearly depicted in the original figure. To address this, we have included a new Supplementary Figure (SI Fig. 6) that explicitly illustrates the N→C parallel-to-antiparallel transition in the native LC. This addition provides a clearer representation of the terminal orientations, enhancing the accuracy and interpretability of our structural data.

We have ensured that all revisions align with the reviewer's suggestions and hope that these modifications meet their expectations. The manuscript has been updated accordingly, and the changes are highlighted in the revised version for easy reference.

8. What is the key step in the misfolding of the natively folded light chain proteins that initiate fibril formation and arrangements into polymorph A and B?

REPLY: We thank the reviewer for this important question. As the reviewer surmises, a key step is the initial formation of a protofilament from the natively folded protein, driven by a combination of factors including mutations, co-factors, and the tissue environment. The subsequent arrangement into distinct polymorphs (A and B) depends on the stability of the intermolecular interactions between these protofilaments. Stable interactions allow them to dimerize (or form higher-order assemblies like trimers), and the specific nature of these interactions dictates the final polymorphic structure.

We have added a description of this key step and its implications for polymorphism to the discussion section to clarify this point.

“.....the renal AL amyloid fibrils exhibit unique topological features. Native LC can form protofilaments under the combined effects of somatic mutations, co-factors, and the local tissue microenvironment. When stable intermolecular interactions exist between protofilaments, two protofilaments can further associate into a dimeric structure, and may even assemble into trimers or higher-order aggregates. Variations in the intermolecular forces can also lead to the formation of distinct polymorphs. The 3D reconstructed fibril is composed of two protofilaments entwined together.....”

Minor points: Check the manuscript for typographical errors – especially in the references.

REPLY: We thank the reviewer for this important reminder. We have carefully proofread the entire manuscript, including the reference list, and corrected all identified typographical and formatting errors.